

# Parameter uncertainty analysis for an operational hydrological model using residual based and limits of acceptability approaches

Aynom T. Tweldebrahn[1], John F. Burkhart[1,2], Thomas V. Schuler[1]

[1]Department of Geosciences, University of Oslo, Oslo, Norway
[2]Statkraft, Oslo, Norway

*Correspondence to*: Aynom T. Tweldebrahn (aynomtt@geo.uio.no)

**Abstract.** Parameter uncertainty estimation is one of the major challenges in hydrological modelling. Here we present parameter uncertainty analysis of a recently released distributed conceptual hydrological model applied in the Nea catchment, Norway. Two variants of the generalized likelihood uncertainty estimation (GLUE) methodologies, one based on the
residuals and the other on the limits of acceptability, were employed. Streamflow and remote sensing snow cover data were used in conditioning model parameters and in model validation. When using the GLUE limit of acceptability (GLUE LOA) approach, a streamflow observation error of 25 % was assumed. Neither the original limits, nor relaxing the limits up to a physically meaningful value, yielded a behavioural model capable of predicting streamflow within the limits in 100 % of the observations. As an alternative to relaxing the limits; the requirement for percentage of model predictions falling within the
original limits was relaxed. An empirical approach was introduced to define the degree of relaxation. The result shows that snow and water balance related parameters induce relatively higher streamflow uncertainty than catchment response parameters. Comparable results were obtained from behavioural models selected using the two GLUE methodologies.

## 1   Introduction

Hydrological models have numerous applications of central importance to society including for planning, design, and
management of environmental and water resources. The operation of hydropower systems is mainly constrained by the availability of water resources. Hydrological models play an important role in forecasting the local inflows to the system on scales ranging from hours to years. With due recognition to the need for accurate prediction of streamflow and snow storage, Statkraft has recently released a new modelling framework mainly tailored for an operational purpose. In this study, one of the conceptual models of this framework was subjected to uncertainty analysis. Conceptual hydrological models typically
have one or more calibration parameters and commonly require some form of inverse modelling to estimate model parameters from observations (Crawford and Linsley, 1966). During calibration, equifinality arises when different parameter sets are equally good at reproducing an output signal (Beven, 1993; Savenije, 2001). The generalized likelihood uncertainty estimation (GLUE) methodology (Beven and Binley, 1992) is an extension of the generalized sensitivity analysis concept of Hornberger and Spear (1981); and it accepts equifinality as a working paradigm for parameter calibration of hydrological
models (Choi and Beven, 2007). It is based on the concept that all models of hydrological systems are highly simplified representations of reality (e.g. Reichert and Omlin, 1997), and hence the selection of a single model as the 'true' description of the complex system is less justifiable. Instead, it is typical to have several different model structures and parameter sets that describe the system in an adequate way (Blazkova and Beven, 2002).

Hydrological modelling is affected by four main sources of uncertainty related to input data, validation data, model
structure, and model parameters (e.g. Renard et al, 2010). Input data uncertainties may arise from measurement limitations and scaling issues, for example, due to forcing data downscaling. Errors of the rating curve affect streamflow estimates and



thereby lead to validation data uncertainty. Structural uncertainty may result from the underlying assumptions and simplifications in the model formulation as well as from application of the model to conditions inconsistent with the model structure (Tripp and Niemann, 2008). Parametric uncertainty reflects the inability to specify exact values of model parameters (Renard et al, 2010) and it may stem from errors in input data and observations used for model conditioning as well as due to epistemic errors in model structure. An increased awareness of these modelling uncertainties and the need for quality control of such models requires the integration of uncertainty analysis into the modelling process from the very beginning (Beven, 1989; Saltelli et al., 2006, Refsgaard et al., 2007).

Uncertainty analysis techniques can be classified as frequentist or Bayesian approaches, probabilistic or non-probabilistic approaches (e.g. Montanari et al. 2009), or as formal or informal approaches (e.g. Vrugt et al., 2009). Among the most widely used techniques in hydrological modelling are the formal Bayesian and the GLUE methods (Jin et al., 2010). The formal Bayesian approach makes strong assumptions about the statistics of observed data; with the likelihood function defined based on assumptions about the nature of the residuals (Schoups and Vrugt, 2010). However, the choice of an adequate likelihood function has been the subject of considerable debate. According to Beven and Smith (2014), a formal probabilistic likelihood function will have limited value since non-stationary epistemic uncertainties cannot be adequately represented by a statistical model. In GLUE, the likelihood measure is associated with a parameter set and should ideally reflect all the different sources of uncertainty (Beven and Smith, 2014). The original GLUE methodology has been subject of debate for using a subjectively set threshold of behavioral models. This problem is common to most residual-based model selection methods (Schaefli, 2016). The extended concept of behavioral models in the GLUE limits of acceptability approach (GLUE LOA) (Beven, 2006) attempts to overcome this drawback through use of error bounds of the observational dataset.

The GLUE LOA methodology involves specifying limits around some observational data within which model predictions are required to lie and thereby considered acceptable for the intended model application. The acceptability limits are set prior to running a model and, among other considerations, they are expected to take into account incommensurability and uncertainty in both the input and evaluation data (Beven, 2009). However, identification of models that reproduce the observed system behavior within the limits of measurement error is not easy due to time-varying errors in the input data and model structure (e.g. Beven, 2016). This difficulty is even more pronounced when input and other sources of errors are not explicitly accounted in defining the LOA.

Good quality time series data and associated uncertainties are not always readily available. For example, in regulated catchments the inflow hydrograph is often estimated from changes in storage volume and outflows using the water balance equation. Thus, as in the case of our study catchment, no stage - discharge relationship exists for estimating the streamflow uncertainty using the usual practice, i.e. by fitting different rating curves. In such instances the alternative is to assume an observation error proportional to the observational data. However, the identification of behavioral models without due consideration to such less precise observation error estimates may lead to the rejection of a useful model (i.e. making Type II error). Some of the measures taken to minimize the risk of making type II error when identifying behavioral models using the GLUE LOA include: extending the limits (e.g. Blazkova and Beven, 2009; Liu et al., 2009) as well as using different model realization for different periods of a hydrological year (e.g. Choi and Beven, 2007). In this study, instead of relaxing the limits, the percentage of observations where model predictions are required to fall within the acceptability limits was relaxed.

The GLUE methodology has been widely used in various disciplines (Beven, 2009; Efstratiadis and Koutsoyiannis, 2010) primarily due to its conceptual simplicity and ease of implementation. Further, its suitability for parallel implementation on





distributed computer systems as well as its general strategy in dealing with equifinality in model calibration make it an appealing framework (Blasone et al., 2008; Shen et al., 2012; Mirzaei et al., 2015).

In this study model parameters were constrained using streamflow and the MODIS snow cover product (Hall et al., 2006). Multi-criteria model conditioning helps to reduce prediction uncertainty through improved parameter identification (e.g.
Efstratiadis and Koutsoyiannis, 2010; Finger et al., 2015); and GLUE provides a flexible approach for using multi-criteria methods through different ways of combining measures. Besides streamflow, one of the observations commonly used in multi-criteria conditioning of rainfall-runoff models in snow dominated catchments is snow data. Remote sensing snow cover data have been used in several hydrological modelling studies for deriving and updating a snow depletion curve (e.g. Lee et al., 2005; Kolberg and Gottschalk, 2006; Bavera et al., 2012); as well as in multi-criteria based model calibration and
simulated snow cover validation (e.g. Udnaes et al., 2007; Parajka and Bloschl, 2008; Berezowski et al., 2015). However, studies involving combined uncertainty of streamflow and snow cover predictions using the GLUE methodology are still missing in the literature.

The main objective of this study is to assess parameter uncertainty for a recently developed distributed conceptual hydrological model using the GLUE methodology with due consideration to the model's main application as an operational
hydrological model. The second objective is to investigate the potential value of snow cover data as additional observation in conditioning model parameters in the study area. The third objective is to assess the possibility of using a time relaxed GLUE LOA approach for constraining model parameters. In doing so, we employ a novel empirical approach for implicit accounting for the effects of input and observational data errors by relaxing the percentage of time steps in which prediction of model realizations fall within the limits.

This paper is organized as follows. First the hydrological model as well as the study site and relevant data used in this study are briefly described in sections 2.1 and 2.2. The procedures followed to setup the uncertainty analyses are then outlined in section 2.3. In section 3, the results from parameter uncertainty as well as uncertainty of streamflow and snow cover predictions using the residual based GLUE approach are presented. The results from the relaxed GLUE LOA are also presented in this section. Finally in sections 4 and 5, the analysis results and their implication on the hydrologic model, the
data as well as the methodologies followed are discussed and conclusions are drawn.

## 2 Methods and materials

### 2.1 The hydrological model

The Statkraft Hydrological Forecasting Toolbox, Shyft, (https://github.com/statkraft/shyft) is an open-source distributed hydrological modelling framework developed by Statkraft (Burkhart et al., 2016). The modelling framework has three main
models (method stacks) and in this study, the PT_GS_K model was used for uncertainty analysis. PT_GS_K uses the Priestley-Taylor (PT) method (Priestley, 1972) for estimating potential evaporation; a quasi-physical based method for snow melt, sub-grid snow distribution and mass balance calculations (GS method); and a simple storage-discharge function (Lambert, 1972; Kirchner, 2009) for catchment response calculation (K). Overall, these three methods constitute the PT_GS_K model in Shyft. The framework establishes a sequence of spatially distributed cells of arbitrary size and shape. As
such it can provide lumped (single cell) or discretized (spatially distributed) calculations, as in this study. The model was applied to each of the grid-cells and for each time step.

Within the GS method, precipitation falling in each grid-cell is classified as solid or liquid precipitation depending on a threshold temperature ($tx$) and on the local temperature values. The snow melt energy is the sum effect of different energy





sources in the system such as short wave and long wave radiation as well as the turbulent sensible and latent energy fluxes. Among other factors, the energy contribution from short wave radiation depends on snow albedo. For a given time step ($t$), the snow albedo of each grid cell depends on the minimum ($\alpha_{min}$) and maximum ($\alpha_{max}$) albedo values as well as on air temperature ($T_a$) (Eq. 1). In this method the decay rates of albedo due to snow ageing as a function of temperature, i.e. the

fast (*fast ADR*, $\alpha_{fdr}$) and slow (*slow ADR*, $\alpha_{sdr}$) albedo decay rates corresponding to temperature conditions above and below $0^oC$ respectively, are parameterized. Turbulent heat contribution is the sum of latent and sensible heat. Wind turbulence is linearly related to wind speed using a *wind constant* and *wind scale* from the intercept and slope of the linear function, respectively (Hegdahl et al., 2016).

$$\alpha_t = \begin{cases} \alpha_{min} + (\alpha_{t-1} - \alpha_{min}) . \left(\frac{1}{2^{1/\alpha_{fdr}}}\right) & T_a > 0 \;^oC \\ \alpha_{t-1} + (\alpha_{max} - \alpha_{min}) . \left(\frac{1}{2\,(\alpha_{sdr})}\right) & T_a \leq \; 0 \;^oC \end{cases} \qquad (Eq.\,1)$$

Sub-grid snow distribution is described by a three parameter Gamma probability distribution snow depletion curve (SDC)

(Liston, 1999; Kolberg and Gottschalk, 2006). The traditional Gamma distribution is parameterized with two values, i.e. the average amount of snow at the onset of the melt season $m$ (mm) and the shape value ($k$); based on the assumption that the ground is completely snow covered before the onset of melt. Since this assumption may not hold true for a number of grid cells especially in alpine areas, a third parameter representing the bare ground fraction at the onset of snow melt season has been introduced (Kolberg and Gottschalk, 2006). The two parameter Gamma distribution (Eq. 2) is thus applied only to the

remaining portion of a grid cell to estimate the fraction of the initially snow covered area where snow has disappeared (y'). The initial bare ground fraction parameter is constant for all years. At each time step, the state parameters such as snow water equivalent (SWE) and snow cover area (SCA) are updated using the SDC function. In the GS method, the shape value is a direct transformation of the sub-grid snow coefficient of variation ($CV_s$).

$$y' = \int_0^{\lambda(t)} f(x; k, \theta) dx = \gamma(k, \tfrac{\lambda}{\theta}) \qquad (2)$$

Where $f$ denotes the Gamma probability density function and $\gamma$ is the incomplete Gamma function. $x$ and $\lambda(t)$ respectively refer to point snow storage and the accumulated melt depth (mm) at time t since the onset of the melt season. $\theta$ represents the scale parameter with $m = k\theta$ and $k = CV_s^{-2}$.

The catchment response function is based on the storage-discharge relationship concept described in Kirchner (2009) and represents the sensitivity of discharge to changes in storage (Eq. 3). This method is based on the idea that catchment

sensitivity to changes in storage i.e. g(Q) can be estimated from the time series of discharge alone through fitting empirical functions to the data such as the quadratic equation. Since discharge is generally non-linear and typically varies by many orders of magnitude, the recommended approach is to use log transformed discharge values in order to avoid the risk of numerical instability. In this method, the three parameters of the catchment response function, i.e. *c1*, *c2*, and *c3* are parameterized.

$$\frac{d(\ln(Q))}{dt} = g(Q)\left(\frac{P-E}{Q} - 1\right) \qquad (3)$$

with, $g(Q) = e^{c1 + c2(\ln(Q)) + c3(\ln(Q))^2}$

In which E and Q respectively represent actual evapotranspiration and discharge. In the original formulation P refers to precipitation, whereas in this method it refers to the liquid water supply from rainfall and snow melt.

The potential evaporation calculation in the PT method requires net radiation and the slope of saturated vapor pressure as

well as the Priestley-Taylor parameter, the psychometric constant, and the latent heat of vaporization (e.g. Matt et al., 2018).





The latter three variables are kept constant in the PT method. Actual evapotranspiration is assumed to take place only from snow free areas and it is estimated as a function of potential evapotranspiration and a scaling factor.

In the default parameter settings of the PT_GS_K model seven parameters are allowed to vary in conditioning the model. The same setting was also followed in this study with the addition that the sub-grid snow coefficient of variation was also

considered an uncertain model parameter. Table 1 shows list of these parameters with their range of possible values.

## 2.2 Study area and data

This study was conducted using climatic and catchment data from the Nea-catchment (11.67390 ° - 12.46273 ° E, 62.77916 ° - 63.20405 ° N). The Nea-catchment constitutes the headwaters of the Nea-Nidelva water resources management area which is situated in Sør-Trøndelag County, Norway. The hydropower generated from this area is the main source of electric supply

to several places in mid-Norway including to one of the biggest cities in the country, Trondheim. As a result this area has significance for Statkraft AS and other stakeholders responsible for the development and management of water resources in the region; and has been selected for research focused on better prediction and understanding of the snow processes and their impact on hydrology of the downstream area.

The Nea-catchment covers a total area of 703 km$^2$ and it is characterized by a wide range of physiographic and land cover

characteristics. Altitude of the catchment ranges from 1783 masl on the eastern part around the mountains of Storsylen to 649 masl at its outlet around the Nea-bru hydrological station. Mean annual precipitation for the hydrological years 2011-2014 was 1120 mm. The highest and lowest average daily temperature values for this period were 28 °C and -30 °C, respectively.

The PT_GS_K model requires temperature, precipitation, radiation, relative humidity, and wind speed as forcing data. In

this study, daily time series data of these parameters for the study area were obtained from Statkraft as point measurement, with the exception of relative humidity. Daily gridded relative humidity data was retrieved from ERA-interim (Dee et al., 2011). The Model uses a Bayesian Kriging approach to distribute the point temperature data over the domain, while for the other forcing variables it uses an inverse distance weighting approach.

Two observational datasets, streamflow and snow cover, were used in this study. Daily observed streamflow

measurements covering four hydrological years (September 1 to August 31) were provided for the study area. The climatic data show that these hydrological years represented periods both above and below the long-term average annual precipitation. Years 2011 and 2013, respectively, were the wettest and driest years in over 10 years. Daily snow cover fraction data (SCF) was retrieved from NASA MODIS snow cover products (MODIS SCF) (Hall et al., 2006). Frequent cloud cover is one of the major challenges when using MODIS and other optical remote sensing data in Norway. In order to minimize the effect of

obstructions and misclassification errors emanating from clouds and other sources, a composite dataset was formed using data retrieved from the Aqua and Terra satellites, MYD10A1 and MOD10A1 products respectively

In this analysis, PT_GS_K was setup in distributed mode over 812 grid cells; requiring the following physiographic data of each grid cell: average elevation, and grid cell total area, as well as the areal fractions of forest, reservoir, lake, and glacier. Data for these physiographic variables was retrieved from two sources: the land cover data from copernicus land monitoring

service (https://land.copernicus.eu/pan-european/corine-land-cover) and the 10m digital elevation model (10m DEM) from the Norwegian mapping authority (Kartverket.no).



### 2.3 The uncertainty analysis methods

In this study modelling and parameter uncertainty analysis was conducted using two GLUE variants. First, the hydrological model and its snow sub-model were subjected to uncertainty analysis using the residual based GLUE methodology. When using this approach, the relevant model parameters were initially conditioned using either streamflow or MODIS SCF. In subsequent analysis, they were conditioned using both streamflow and SCF. Following that, uncertainty analysis was conducted using the relaxed GLUE LOA approach.

### 2.3.1 Sampling the parameter dimensions

The performance of all uncertainty analysis techniques depends on the efficiency of the sample to represent the entire response surface (Pappenberger et al., 2008). In this study, prior distributions of the uncertain model parameters were not known and hence a uniform distribution was assumed. The challenge in using uniform distribution is, however, to adequately sample the entire parameter dimensions. To overcome this challenge and to better identify regions of behavioral simulations, a sample size of 100,000 runs was used. Each model run is a realization of a parameter set randomly drawn from the domains of the model parameters. Matlab scripts from the SAFE toolbox (Pianosi et al., 2015) were used as a basis to characterize behavioral and non-behavioral models.

### 2.3.2 The residual based GLUE approach

In this study, the performance of each model realization was evaluated by using relevant likelihood measures. Residual based informal likelihood measures are considered suitable measures of fit when large data sets such as rainfall-runoff time series exist for model conditioning (Hassan et al., 2008). The Nash-Sutcliffe efficiency (NSE, Eq. 4) belongs to these groups of likelihood measures; and it is the most widely-used likelihood measure for assessing the fitness of model parameters in hydrological modelling (Xiong and O'Connor, 2008). Further the main end users of the model commonly use NSE both in calibration and evaluation of hydrological models. Thus use of this performance measure as a streamflow likelihood measure makes it easier both in setting the threshold value for behavioral models (i.e. based on previous experience) and in communicating model performance outputs. However, the NSE calculated using raw values tends to overestimate model performance during peak streamflow and underestimate during low streamflow conditions (e.g. Krause et al., 2005). To partly overcome this problem, NSE is often calculated with log transformed observed and simulated values. In this study, both NSE and NSE with log transformed streamflow values (LnNSE) were thus employed as likelihood measures in evaluating each model run.

$$NSE = 1 - \frac{\sum_{i=1}^{n}(Q_{sim,i} - Q_{obs,i})^2}{\sum_{i=1}^{n}(Q_{obs,i} - \bar{Q}_{obs})^2} \tag{4}$$

In which $Q_{sim}$ represents simulated streamflow, $Q_{obs}$ is observed streamflow and $\bar{Q}_{obs}$ represents mean value of observed streamflow series.

Within the residual based GLUE procedure, the definition of threshold likelihood value at which the model performance is judged reasonable is a subjective choice by the modeler. In this study, NSE and LnNSE of 0.7 and 0.6 were respectively considered as the threshold values for behavioral models. These values were chosen with due consideration to the input and observational data quality as well as the relative importance given to high streamflow in relation to low streamflow conditions in the hydropower industries. In the case of the combined likelihood measure, a weighted average threshold value



(e.g. Hassan et al., 2008) was calculated assuming each likelihood measure to have a weight proportional to its threshold value. Accordingly, the NSE and LnNSE likelihood measures were respectively assigned weights of 0.54 and 0.46 (Eq. 5).

$$L\big(O\big|M(\theta_i)\big) = 0.46(L_{LnNSE}) + 0.54(L_{NSE}) \tag{5}$$

where $L\big(O\big|M(\theta_i)\big)$ represents the combined likelihood measure for the i[th] model realization with model prediction of $M(\theta_i)$

which is a function of the set of model parameters $\theta_i$, and corresponding to the observations $(O)$. $L_{NSE}$ and $L_{LnNSE}$ respectively represent the likelihood measures based on NSE and LnNSE. Models producing likelihood measure values greater than or equal to the threshold value were labeled as behavioral models and were retained for use in further analysis.

The root mean squared error (RMSE) of simulated and MODIS fractional snow cover was used as a likelihood measure of SCF. A threshold value of 0.17 was set when using the RMSE in model conditioning. This value was fixed based on

average performance of similar conceptual hydrological models as a reference (e.g. Skaugen and Weltzien, 2016); and with due consideration to the inherent error in the MODIS SCF data. The estimated annual average error of MODIS SCF maps for Northern Hemisphere is approximately 8 % in the absence of cloud (Pu et al., 2007); and in forest dominated areas it may reach up to 15 % (MODIS, 2010).

Preliminary assessment of model performance indicates that the snow yes/no based model performance (CSI, Table 2) is

very high both before the onset of snow melt and during the complete melt out period. The lowest mismatch between simulated and MODIS SCF was observed during early summer. It was thus decided to use a weighted mean likelihood measure of SCF; with maximum weight assigned to likelihoods from the mid part of the observation period. The likelihood of each SCF observation was assigned a specific weight based on the location of the observation date in a trapezoidal membership function (TMF). The start and end of MODIS SCF observation period locate the feet of the trapezoid and the

start and end of the month of June locate the shoulders (Fig. 1). For each model realization, the weighted average RMSE (wRMSE) of all SCF observations and their corresponding simulated values for the calibration period was calculated and model realizations with wRMSE below the threshold value were considered behavioral. The weight of each behavioral model was calculated as the inverse of wRMSE and was used in constructing the cumulative distribution function (CDF); based on which the predicted SCF values for different quantiles can be extracted.

When selecting behavioral models using the combined likelihoods of streamflow and SCF, the merging of these likelihoods was carried out in two steps. First the likelihoods representing low and high flow condition, *viz.* LnNSE and NSE were combined following similar procedure as described above. The likelihoods of streamflow and SCF were separately rescaled such that their respective weights would sum to unity following a similar procedure to that used in Brazier et al. (2000). The combined streamflow likelihood and the SCF likelihood were subsequently multiplied to get a combined

likelihood measure of streamflow and SCF.

### 2.3.3 The relaxed GLUE LOA approach

Unlike the residual based model selection approaches, including the residual based GLUE methodology, the GLUE LOA approach relies on an assessment of uncertainty in the observational data. Uncertainty analysis was also thus conducted in this study using the GLUE LOA approach and its results compared against those from the residual based GLUE

methodology.

In this study when using the GLUE LOA approach, both the streamflow and MODIS SCF data were considered as uncertain observations. Since no uncertainty data was available for streamflow observations in the study site, mean streamflow uncertainty of 25 % was assumed and the streamflow limits were defined using this value. Although, the maximum expected error of MODIS snow cover products under clear-sky conditions is reported to be 15 % for forest areas



(MODIS, 2010), cloud coverage coupled with lack of contrast between clouds and snow cover may severely affect the accuracy. And in some cases this leads to misclassification of snow as land (e.g. Parajka et al., 2012). Thus a SCF uncertainty of 25-50 % was assumed to represent the errors associated with the SCF observations and the input data.

An alternative approach was employed to minimize the risk of rejecting useful model realizations due to using assumed average observational error bounds and due to lack of explicitly accounting the time-varying level of observational and input data uncertainties. The procedure involves relaxing the percentage of observations where model predictions fall within the acceptability limits. Model realizations whose predictions fall within the acceptable bounds in a defined percentage of the observations were considered behavioral. The minimum acceptable percentage of observations where model predictions fall within the limits (hereafter referred as threshold pLOA) in turn was set such that the 5-95 % prediction limit of streamflow,

reported as Containing Ratio (CR, see Eq. 5), is close to the value obtained using the residual based GLUE methodology. The procedure for relaxing the original GLUE LOA requirement during the calibration period involves the following steps:

**Step 1:** define an acceptable prediction limit (CR) at a chosen certainty level (e.g. 5-95 %). In this study the CR value obtained for the calibration period using the residual based GLUE methodology was adopted as an acceptable CR value.

**Step 2:** relax the acceptable percentage of observations where model predictions fall within the limits. This is done by

gradually lowering the requirement for bracketing the observations in 100% of the time steps up to the acceptable pLOA.

**Step 3:** run calibration and test whether each model realization prediction falls within the limits at least for the specified percentage of the total observations. If model realizations that satisfy the relaxed acceptability criteria are found, proceed to step 4, otherwise lower the threshold pLOA further and repeat this step.

**Step 4:** calculate the new CR and check if it is close to the predefined acceptable CR value. If the calculated CR is less than

the predefined CR, repeat steps 2 to 4. Whereas, if the two CR values are close (e.g. within 5%) then accept all model realizations that satisfy this pLOA as behavioral and store their indices for use in further analysis.

    Model realizations that fulfill this relaxed LOA criteria both in streamflow and SCF observations were considered behavioral. A triangular membership function was used to define the weights of each criterion, where a maximum weight of

1.0 was assigned to predictions with a perfect match to the observation and a minimum weight of 0.0 to predictions outside the acceptability limits. For each model realization, the weights of individual time-steps were added to give a generalized weight. Following the procedure by Blazkova and Beven (2009), the weights associated with streamflow and MODIS SCF were combined by taking the sum of these two criteria and rescaling them such that the sum of the weights for behavioral models is unity. The behavioral model realizations were used for prediction weighted by their overall degree of performance.

**2.3.4 GLUE output analysis**

    A split-sample based cross-validation of streamflow predictions was used to alternately evaluate how well the behavioral models identified at a given calibration period are able to reproduce the observed values from another period. The hydrologic model was run for four years at daily time step. The first month of each hydrological year was considered as a spin up period; and hence excluded from all uncertainty analyses. Each of the four years was alternately used to identify behavioral models

and the remaining three years were individually used to assess the modelling uncertainty.

    In this study the modelling uncertainty was evaluated using both qualitative and quantitative evaluation techniques. The upper and lower streamflow prediction limits as well as observed values were plotted on the same graph to visually assess capability of the identified behavioral models in bracketing the observations. The Containing Ratio (CR) index was also used to analyze the prediction uncertainty following a similar procedure to that used in some studies involving the GLUE





methodology (e.g. Xiong et al., 2009; He et al., 2011). CR is expressed as the ratio of the number of observations falling within respective prediction bounds to the total number of observations (Eq. 5).

$$CR = \frac{\sum_{i=1}^{n} I(Q_{obs,i})}{n} \tag{5}$$

where: $I(Q_{obs,i}) = \begin{cases} 1, & L_{lim,i} < Q_{obs,i} < U_{lim,i} \\ 0, & Otherwise \end{cases}$

$Q_{obs,i}$ represents observed streamflow at the i[th] time step; and $L_{lim,i}$ and $U_{lim,i}$ are the lower and upper prediction bounds respectively.

As an alternative to a crisp prediction for an observation (e.g. Xiong and O'Connor, 2008), the median (50 %) streamflow prediction was also estimated from the behavioral model simulations and compared against observations using both NSE and LnNSE as goodness of fit measures. Similarly, the critical success index (CSI, Table 2) and RMSE were used

as goodness of fit measures for median SCF prediction. When using RMSE, the fractional snow cover data of each grid cell was directly employed in validating median predictions. CSI represents the number of grid-cells where the snow events are correctly predicted out of the total number of grid-cells where snow is predicted in the model. It was calculated based on a binary snow cover data using the two by two contingency table analysis (Table 2) following a similar procedure to that used in Hanzer et al. (2016). When converting the snow cover fraction to a binary measure, a grid-cell was classified as snow

covered if at least 50 % of its area is snow covered.

## 3   Results

### 3.1 Uncertainty analysis using the residual based GLUE approach

#### 3.1.1   Uncertainty of model parameters

The uncertainty of model parameters was analyzed using all years of record together as a single time series data. The dotty
plots (Fig. 2) depict the goodness of fit response surface projected onto individual parameter dimensions. The parallel coordinate plots (Fig. 3) also show the distribution of model parameters within their respective parameter dimensions. The distribution of behavioral simulations across a parameter dimension varies from one parameter to another. The behavioral models are scattered nearly across the entire range of parameter dimension for *fast ADR*, *slow ADR*, and *snow CV*; indicating low model sensitivity to these parameters. On the other hand, the relatively localized distribution of behavioral models
towards lower values when projected onto the parameter ranges of *c1*, *c2*, *tx*, and *wind scale* as well as towards higher values of *c3* reflects higher sensitivity of simulated streamflow to these calibration parameters. Furthermore, the parallel coordinate plots show an increase in likelihood measure value towards the lower (for *c1*, *c2*, *tx*, and *wind scale*) and higher (for *c3*) parts of their respective parameter dimensions.

The aforementioned less sensitive model parameters can, however, have high effect on model outputs through interaction
with other parameters. Some degree of interaction between model parameters can be seen from the correlation shown in Fig. 4. For example, a general decreasing trend in model performance can be noticed with a joint increase in *c1* and *c2*. The strong influence of *tx* in constraining the output is also evident in these plots.

The posterior distribution histograms (Fig. 5) and the statistical summary table of posterior distribution (Table 3) illustrate variability in distribution characteristics of the model parameters. The catchment response parameters, *viz. c1*, *c2*,
and *c3* showed relatively well defined peaks. Whereas, *fast ADR*, *slow ADR*, and *snow CV* appear less identifiable with





relatively flat distribution across their respective parameter dimensions. It should however be remembered that in the GLUE methodology, it is the set of parameter values that gives a behavioral model.

### 3.1.2 Uncertainty of streamflow predictions

Figure 6 shows a sample cross-validation of daily streamflow prediction limits against observed values. The upper and lower
prediction bounds as well as the median values are generated with behavioral models identified in year 2011 using the combined NSE and LnNSE likelihood measure. The calculated uncertainty in streamflow prediction indicated by the 5-95 percentile range (shaded band) varied over time and relatively higher uncertainty was noticed during high streamflow than low streamflow periods.

As can be seen from the summary table of cross-validation results (Table 4), the CR values range from 0.62 to 0.91 with
an overall mean value of 0.77. The mean CR values for the calibration and validation periods are 0.78 and 0.76 respectively. The evaluation result generally shows that the median prediction of behavioral models selected using the combined likelihood was able to reproduce the observed values remarkably well with average NSE and LnNSE of 0.86 and 0.72 respectively for the validation period. However, performance of the behavioral models identified using NSE was very low when evaluated using LnNSE in year 2014.

### 15   3.1.3 Uncertainty of snow cover predictions

Snow cover fractions (SCF) and snow water equivalent (SWE) are two main outputs of the snow sub-model (GS) of the PT_GS_K model. In this study an initial single-likelihood based conditioning of the GS specific parameters was carried out using MODIS SCF only and RMSE as a measure of model performance.

The cross-validation result of predicted median values against MODIS SCF observations is shown in Table 5. The
highest and lowest RMSE values during the calibration period were 0.15 and 0.06 respectively with an average RMSE value of 0.11. Minimum and maximum RMSE values of 0.06 and 0.22 respectively were observed during the validation period with an average RMSE value of 0.13. Similarly the lowest CSI during the calibration and validation periods were 0.99 and 0.88 respectively. Comparable maximum CSI results were observed between the two periods. The 5-95 % SCF prediction interval was able to reasonably bracket the observations in most of the calibration and validation periods with mean CR
values of 0.60 and 0.71 respectively without any explicit accounting for model residuals for each parameter set. The inter-annual comparison of model performance shows that relatively lower performance was observed in years 2011 and 2012 as compared to the other periods.

### 3.1.4 Uncertainty of streamflow and snow cover predictions using both observations

The cross-validation result of simulated streamflow and SCF against observations is shown in Table 6.  A similar model
performance was observed when model parameters are conditioned using both streamflow and MODIS SCF as compared to when only streamflow was used for model conditioning. The mean NSE and LnNSE values of the median streamflow prediction in the validation periods were 0.85 and 0.71 respectively. The average streamflow prediction uncertainty (CR) in the validation period was 0.70. For SCF, an average RMSE and CSI values of 0.11 and 0.99 respectively were obtained when using the combined likelihood. The streamflow and SCF median predictions obtained in this analysis are similar to the
results when model parameters are respectively conditioned with streamflow only or MODIS SCF only. This result shows that contribution from the information content of MODIS SCF was less significant in constraining the model parameters.



The relatively low quality of MODIS SCF data as compared to the streamflow data for the study site may also partly explain this phenomenon.

### 3.2 Uncertainty analysis using the relaxed GLUE LOA approach

The median streamflow prediction of behavioural models identified using the relaxed GLUE LOA was able to mimic the observed values very well with a mean NSE and LnNSE of 0.85 and 0.7, respectively for the validation period (Table 7). A comparable performance was observed between models selected using the residual based GLUE and the relaxed GLUE LOA. The similarity in median predicted streamflow by these two GLUE methodologies can also be noticed from visual comparison of the resulting hydrographs (Fig. 6 and Fig. 7). A mean streamflow CR value of 0.75 was obtained for the validation period when using the relaxed GLUE LOA. This shows slightly better capability of the 5-95 % prediction bounds

in bracketing the observations as compared to predictions using the residual based GLUE methodology when both streamflow and SCF are used in model conditioning.

The behavioural models selected using the relaxed GLUE LOA approach were also able to adequately reproduce observed SCF with a mean RMSE and CSI of 0.11 and 0.98 respectively for the validation period. Generally high prediction uncertainty of SCF was observed during the onset of snow melt and low uncertainty during the summer with an average CR

of 0.63. Thus hydrological year 2011, having most of its observations coming from April, showed the lowest CR as compared to the other periods. Figure 8 shows observed and simulated average catchment SCF for sample calibration period (2011) and validation period (2012). From this figure it can be noticed that the median prediction tends to overestimate the observed SCF values; and many of the observed values from the month of April fall outside the 5-95 % prediction bounds. The overall result, however, indicates an improved capability of the 5-95 % prediction bounds in bracketing the SCF

observations as compared to predictions using the residual based GLUE methodology.

### 4    Discussion

The streamflow prediction uncertainty analyses results show that model performance was relatively lower during low-streamflow than high-streamflow conditions throughout most validation periods (e.g. Table 4). A similar result was reported by Choi and Beven (2007) in their multi-period cluster based uncertainty analysis in Bukmoon catchment, South Korea,

where high percentage of simulation bias was observed during the drier seasons due to relatively poor model performance during these periods. The result of this study is thus consistent with the general observation that catchment hydrologic models perform relatively well in wet conditions, but break down during low streamflow conditions (e.g. Kirchner, 2009). In the case of results from the residual based GLUE methodology, this can also be partly attributed to the nature of the likelihood measure used to identify the behavioral models. The result reveals this observation, where model performance

during low streamflow periods (LnNSE) was improved when using the combined likelihood measures as compared to using NSE alone. This is because models identified using NSE alone strongly reflect hydrologic characteristics of the high streamflow periods and are expected to perform more poorly during low streamflow conditions.

In order to assess the potential value of MODIS SCF in constraining model parameters, the snow sub-model parameters were constrained using this observation and the posterior distribution of the individual parameters were compared against

corresponding distributions that resulted from model conditioning using streamflow only. Parameter inference based on SCF only as a conditioning observation gave some parameter estimates that deviate significantly from those obtained when conditioned with streamflow only (Fig. 9). The box plots depict posterior distribution of the snow related parameters



separately conditioned using streamflow and SCF. For the ease of comparison, parameter values were scaled between 0 and 1. From these plots it can be seen that *tx* and *wind scale* are the model parameters most sensitive to the conditioning data type with a significant shift of their quartiles towards upper part of the parameter dimensions when conditioned using SCF. Whereas the *fast ADR*, *slow ADR*, and *snow CV* did not show significant displacement in their posterior distribution. These

parameters were also identified as the least sensitive model parameters when the model was constrained using streamflow only.

Generally in snow models with the sub-grid snow distribution component parameterized using statistical probability distribution function, low *snow CV* results to faster depletion rate of snow covered fraction (e.g. Liston, 2004). Thus the slight displacement of *snow CV* posterior values towards lower part of its parameter dimensions coupled with the increased

posterior values of *wind scale* would give rise to lower snow cover fraction during the melting period when model parameters are constrained using SCF only. On the other hand, the increased posterior values of rain/snow threshold (*tx*) would result to increase in snow deposition and thereby to partial or full cancelling out of the effects of changes in *snow CV* and *wind scale*. This phenomenon may thus lead to equifinality, where different set of model parameters give comparable SCF responses.

In the GLUE LOA approach a particular model realization is classified as acceptable if its prediction falls within the limits for all observed values. In continuous rainfall-runoff modelling it is difficult for all predictions of a given model realization to lie within the observation limits in a time series. In some cases this phenomenon can be attributed to different specific processes dominating the hydrologic behavior of a catchment at different sub-periods. While in other instances it may be due to lack of a viable means for explicitly taking into account for the effect of variable sources and level of

uncertainties from the input data errors which are difficult to set a priori. Thus the time-varying likely effect of other sources of errors such as input errors on prediction uncertainty need also be implicitly taken into account when defining the limits of acceptability.

The use of GLUE LOA for testing hydrologic models as hypotheses without a due consideration to errors in input data may lead to rejection of useful models that might adequately represent the catchment behavior and thereby to making type II

error (false negative). In the past, various attempts have been made to minimize the risk of making type II errors in model calibration studies using the GLUE and other frameworks. In some studies an improved calibration of hydrologic models was obtained through independent calibration of sub-periods of a time series (e.g. Boyle et al., 2000; Samanta and Mackay, 2003). When it comes to the GLUE LOA approach, extending the limits (e.g. Blazkova and Beven, 2009; Liu et al., 2009) as well as using different model realization for different periods of a hydrological year (e.g. Choi and Beven (2007) are some of

the measures taken to minimize the risk of making Type II errors. Common to all these measures is that they attempt to relax the selection criteria for behavioral models.

In this study when using the GLUE LOA approach, the streamflow bounds were set to +/- 25 % and the result shows that none of the model realizations were able to satisfy the LOA criteria without one or more of their predictions falling outside the acceptable streamflow bounds. The failure rate was higher during low flow conditions as compared to high flow

conditions. An initial attempt was made to relax the limit of acceptability by extending the streamflow bounds. Regardless, no model realization with its predictions falling within the error bounds for all observations was found until the limits were extended to over +/- 85 %. This relaxed acceptability limit seems less reasonable in terms of its physical meaning as an error bound. Therefore, rather than relaxing the limits, an alternative empirical approach was followed by relaxing the number of simulation time steps which fulfilled the original LOA criterion.     The     procedure     involves     defining     the     acceptable





percentage of observations that are required to be bracketed by model predictions (during the calibration period) based on a predefined acceptable CR value.

This empirical approach is based on the observed relationship between prediction uncertainty and number of behavioral models which in turn is a function of the selection criterion. As the threshold value of a likelihood measure increases (in the case of residual based GLUE) or absolute value of the limits decreases (in the case of GLUE LOA), the simulated runoff series gradually converges, though not necessarily to the observations. A similar observation was also reported in other GLUE based uncertainty studies (e.g. Xiong et al., 2008). A further analysis in this study reveals that, as the percentage of observations required to be bracketed by each model realization (pLOA) increases, the number of behavioral models decreases and thereby the simulated runoff series converges resulting to low CR (Figure 10). In this study, the threshold pLOA for each calibration period was defined in such a way that the 5-95 % prediction uncertainties of streamflow using the residual and the LOA based GLUE methodologies are similar. Defining the threshold pLOA this way helps to set a reasonable value that minimizes the risk of making type II errors while maintaining the overall model accuracy by rejecting the inclusion of non-behavioural models. Furthermore, it helps to grossly compare the performance of behavioural models selected using the relaxed GLUE LOA against the residual based GLUE in terms of their ability to reproduce the median streamflow and SCF predictions at similar level of uncertainty (i.e. the CR used to set pLOA).

Although it is difficult to single out the effects of input data error from model structural error on model performance using the GLUE methodology, the error patterns may aid in assessing model performance in different periods of the hydrologic year. Generally, a good model structure coupled with good data is not expected to give a consistent bias (e.g. Liu et al., 2009). Figure 11 shows a sample daily percentage of acceptable simulations satisfying the LOA criteria during the hydrologic year 2012. The percentage of acceptable number of model realizations in each time step was generally low during the calibration period (< 65 %). However, for each time step, predictions from some behavioral models are able to mimic the corresponding observation within the assumed error bound. The percentage of acceptable models was relatively higher during high than low streamflow conditions. And this result is consistent with the general observation that most hydrological models perform relatively well during high streamflow than low streamflow periods. The spike in percentage of acceptable models in the month of February 2012 when time steps around are so low, however, reveals how model performances can unexpectedly vary between time steps in response to input data errors and/or the observational error bounds. The observed spike could thus be attributed to relatively low input data errors and/or lower actual observational error bounds as compared to the assumed average values for the particular time step. The distribution of the behavioral model weights over the calibration period shows that the mean weight during the period where the spike occurred is very low. Similarly the median weight of behavioral models during this period is close to zero implying that most of the model realizations have their predictions that barely fall within the limits.

This result reveals that the GLUE LOA with relaxation in percentage of observations where model predictions fall within observational error bounds can be used as an alternative approach for conditioning model parameters and conducting uncertainty analysis when there is lack of metadata on input and observational data uncertainty coupled with highly time-varying level of uncertainty from such sources. After relaxation, a limited sample of the total observations, i.e. 30 - 40 % of a hydrologic year, was able to effectively identify behavioural models; and this result is consistent with findings of other studies dealing with the effect of observation size on constraining model parameters (e.g. Seibert and Beven, 2009; Liu and Han, 2010; Sun et al., 2017). The information content of the input and observation datasets is more important than the length of the datasets especially in continuous rainfall-runoff modelling.





## 5   Conclusions

Two GLUE methodology variants were applied for parameter uncertainty analysis of a distributed conceptual hydrological model. The analysis result from the residual based GLUE methodology shows that the catchment response parameters, *viz.* *c1*, *c2*, and *c3* as well as the *wind scale* are the most sensitive model parameters. More caution is thus required when

defining the value range of these parameters. On the other hand, the fast and slow albedo decay rates as well as the *snow CV* are relatively more uncertain model parameters.

Model conditioning using combined streamflow and MODIS SCF did not improve the median prediction of streamflow as compared to the result when model parameters are conditioned using streamflow only. A similar result was also observed for SCF predictions. The additional information from the MODIS SCF data was generally less significant in constraining the

rainfall-runoff model parameters.

When using the GLUE LOA approach, the model did not provide any behavioral simulation in the sample tried on the basis of treating the assumed observational error bound as 5-95% error. A relaxation was needed in order to partly overcome the limitations of using constant observational error proportionality and not taking an explicit account of the other sources of uncertainty such as from input data errors. A relaxed GLUE LOA approach was introduced that allows a relaxation on the

number of time steps required to achieve the LOA. Similar results are obtained using both the residual based GLUE and the relaxed GLUE LOA approaches. Relaxing the percentage of observations required to be bracketed per simulation period by a particular model realization (pLOA) was found to be more effective than relaxing the observational error bounds. In this study the 5-95 % prediction uncertainty of residual based GLUE methodology was used as a reference to define the pLOA in the relaxed GLUE LOA analysis using forcing and observational datasets from a single catchment. More similar case studies

should be conducted on catchments with different hydrologic characteristics to assess the scope of this approach under different condition.

*Data availability*. The underlying hydrologic observations for this analysis were provided by Statkraft AS and are proprietary within their hydrologic forecasting system. However, the data may be made available upon request. Please

contact: John Burkhart <john.burkhart@statkraft.com> for further information and access to the data.

*Competing interests*. The authors have no conflict of interest.

*Acknowledgements*. This work was conducted within the Norwegian Research Council's - Enhancing Snow Competency of

Models and Operators (ESCYMO) project (NFR no. 244024) and in cooperation with the strategic research initiative LATICE (Faculty of Mathematics and Natural Sciences, University of Oslo https://mn.uio.no/latice). Computational and data storage resources were provided by NOTUR/NORSTORE projects NS9333K and NN9333K. We are grateful for Keith Beven and Chong-Yu Xu for their helpful comments. Furthermore, we thank Sigbjorn Helset and Statkraft AS, in general, for helping us to setup Shyft in a windows environment and for providing us the data. We also thank Kristoffer Aalstad and

Sebastian Westermann for providing us a matlab script for retrieving the composite MODIS SCF data.

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

**Table 1.** Range of model parameters used for the PT_GS_K model stack uncertainty analysis

| Name | Min. | Max. | Description | Method |
|---|---|---|---|---|
| $c1$ | -5.0 | 1.0 | constant in Catchment Response Function, CRF | K |
| $c2$ | 0.0 | 1.2 | linear coefficient in CRF | K |
| $c3$ | -0.15 | -0.05 | quadratic coefficient in CRF | K |
| $tx$ | -3.0 | 2.0 | Solid/liquid threshold temperature ($^{o}$C) | GS |
| $wind\ scale$ | 1.0 | 6.0 | slope in turbulent wind function | GS |
| $fast\ ADR$ | 1.0 | 15.0 | fast albedo decay rate (days) | GS |
| $slow\ ADR$ | 20.0 | 40.0 | slow albedo decay rate (days) | GS |
| $snow\ cv$ | 0.06 | 0.85 | spatial coefficient of variation of snowfall | GS |

**Table 2.** Set up of the two-by-two contingency table for binary snow cover data comparison. $O$ and $S$ respectively represent observed and simulated binary snow cover and the subscripts refer to snow-free ($0$) and snow covered ($1$) grid cell.

| | $S_1$ | $S_0$ | $Sum$ |
|---|---|---|---|
| $O_1$ | $n_{11}$ | $n_{01}$ | $n_{x1}$ |
| $O_0$ | $n_{10}$ | $n_{00}$ | $n_{x0}$ |
| $Sum$ | $n_{1x}$ | $n_{0x}$ | $n_{xx}$ |

$$CSI = \frac{n_{11}}{n_{xx} - n_{00}}$$

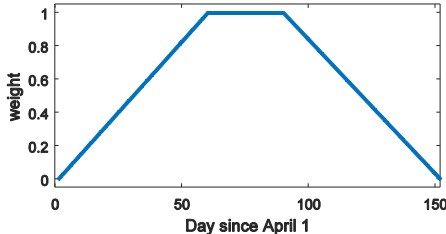

**Figure 1**. A trapezoidal membership function for SCF likelihoods in the observational period.




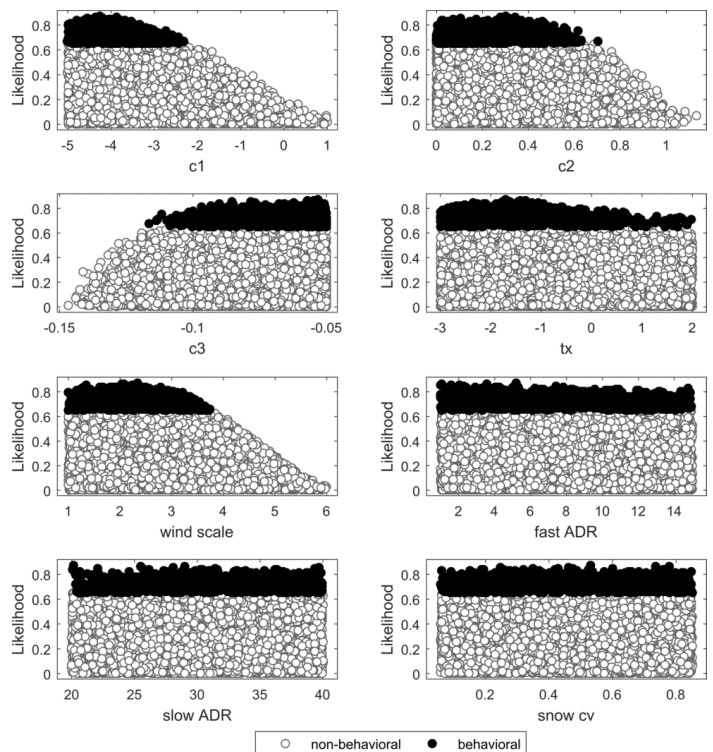

**Figure 2**. Dotty plots of the likelihood measure for behavioral and non-behavioral models identified using the residual based GLUE methodology.

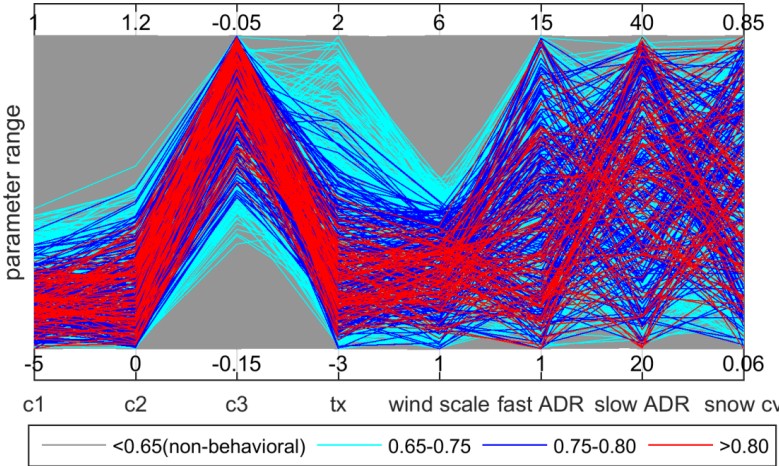

**Figure 3**. Distribution of model parameters within their variability ranges.




**Figure 4**. Model performance in response to the interaction between model parameters.

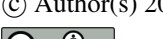

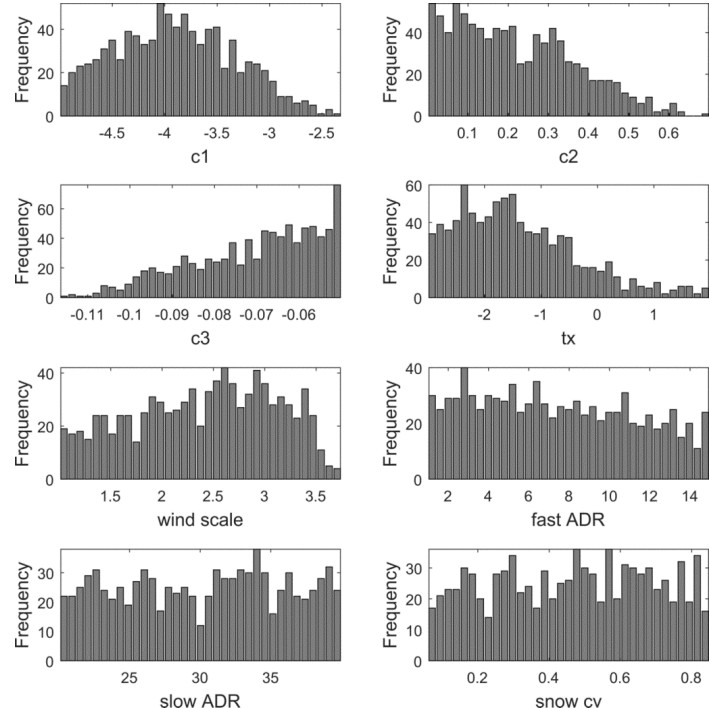

**Figure 5**. Posterior distribution of calibration parameters after conditioning on flow observations

**Table 3**. Statistical summary of posterior distribution for model parameters

| Statistics | c1 | c2 | c3 | tx (°C) | wind scale | fast ADR (days) | slow ADR (days) | snow CV |
|---|---|---|---|---|---|---|---|---|
| Minimum | -5.00 | 0.00 | -0.12 | -3.00 | 1.01 | 1.00 | 20.07 | 0.06 |
| Maximum | -2.32 | 0.70 | -0.05 | 1.98 | 3.74 | 14.96 | 39.98 | 0.85 |
| Mean | -3.90 | 0.22 | -0.07 | -1.39 | 2.40 | 7.38 | 30.21 | 0.46 |
| Median | -3.92 | 0.20 | -0.07 | -1.57 | 2.48 | 7.01 | 30.71 | 0.47 |
| Variance | 0.33 | 0.02 | 0.00 | 1.15 | 0.48 | 15.46 | 33.15 | 0.05 |
| Skewness | 0.18 | 0.53 | -0.58 | 0.81 | -0.22 | 0.19 | -0.05 | -0.06 |

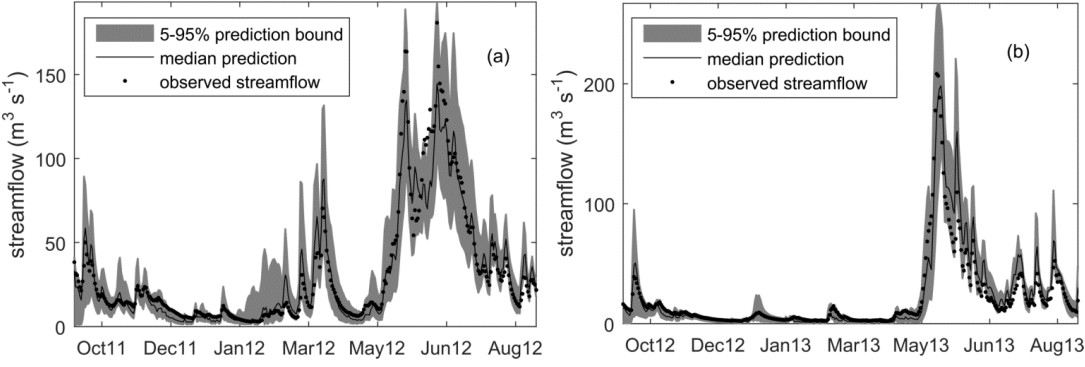

**Figure 6**. Median, 5-95 percentile range and observed values of streamflow for sample calibration period (a) and validation period (b)

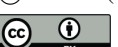



**Table 4.** Cross-validation of streamflow predictions against observed values. Bold numbers show the result for calibration period.

| Valid. year | Likelihood (LH) measure | Calibration year | | | | | | | |
|---|---|---|---|---|---|---|---|---|---|
| | | 2011 | | 2012 | | 2013 | | 2014 | |
| | | NSE | Comb. LH | NSE | Comb. LH | NSE | Comb. LH | NSE | Comb. LH |
| 2011 | NSE | **0.893** | **0.890** | 0.770 | 0.806 | 0.809 | 0.790 | 0.697 | 0.840 |
| | LnNSE | **0.712** | **0.855** | 0.366 | 0.812 | 0.693 | 0.719 | 0.521 | 0.771 |
| | CR | **0.759** | **0.721** | 0.756 | 0.677 | 0.805 | 0.764 | 0.729 | 0.710 |
| 2012 | NSE | 0.842 | 0.869 | **0.920** | **0.930** | 0.818 | 0.787 | 0.910 | 0.874 |
| | LnNSE | 0.753 | 0.878 | **0.694** | **0.890** | 0.640 | 0.616 | 0.685 | 0.792 |
| | CR | 0.885 | 0.844 | **0.866** | **0.844** | 0.907 | 0.882 | 0.852 | 0.803 |
| 2013 | NSE | 0.922 | 0.925 | 0.878 | 0.877 | **0.934** | **0.942** | 0.862 | 0.916 |
| | LnNSE | 0.780 | 0.914 | 0.391 | 0.799 | **0.887** | **0.936** | 0.531 | 0.792 |
| | CR | 0.778 | 0.759 | 0.759 | 0.666 | **0.830** | **0.830** | 0.756 | 0.622 |
| 2014 | NSE | 0.828 | 0.884 | 0.860 | 0.892 | 0.826 | 0.810 | **0.901** | **0.924** |
| | LnNSE | -0.346 | 0.566 | -0.529 | 0.531 | 0.138 | 0.488 | **0.268** | **0.716** |
| | CR | 0.737 | 0.658 | 0.721 | 0.666 | 0.773 | 0.721 | **0.718** | **0.647** |
| No. behavioural models | | 1573 | 749 | 3737 | 1031 | 4725 | 2245 | 4648 | 604 |

**Table 5.** Cross-validation of SCF predictions against MODIS SCF.

| Calib. year | Validation year | | | | | | | | | | | | No. of behav. models |
|---|---|---|---|---|---|---|---|---|---|---|---|---|---|
| | 2011 | | | 2012 | | | 2013 | | | 2014 | | | |
| | RMSE | CSI | CR | RMSE | CSI | CR | RMSE | CSI | CR | RMSE | CSI | CR | |
| 2011 | **0.147** | **0.987** | **0.417** | 0.152 | 0.999 | 0.330 | 0.067 | 0.985 | 0.839 | 0.089 | 0.991 | 0.656 | 83922 |
| 2012 | 0.150 | 0.987 | 0.347 | **0.154** | **0.998** | **0.236** | 0.076 | 0.978 | 0.824 | 0.095 | 0.989 | 0.629 | 84945 |
| 2013 | 0.200 | 0.878 | 0.924 | 0.217 | 0.875 | 0.795 | **0.057** | **0.985** | **0.919** | 0.100 | 0.948 | 0.931 | 98400 |
| 2014 | 0.146 | 0.982 | 0.738 | 0.151 | 0.983 | 0.632 | 0.057 | 0.988 | 0.903 | **0.083** | **0.992** | **0.799** | 95039 |

**Table 6.** Cross-validation of streamflow and SCF predictions.

| Valid. year | Obs. | Likelihood measure | Calibration year | | | |
|---|---|---|---|---|---|---|
| | | | 2011 | 2012 | 2013 | 2014 |
| 2011 | flow | NSE | **0.888** | 0.773 | 0.790 | 0.841 |
| | | LnNSE | **0.856** | 0.780 | 0.711 | 0.769 |
| | | CR | **0.660** | 0.611 | 0.753 | 0.693 |
| | SCF | RMSE | **0.142** | 0.146 | 0.155 | 0.143 |
| | | CSI | **0.987** | 0.987 | 0.954 | 0.987 |
| | | CR | **0.461** | 0.341 | 0.610 | 0.430 |
| 2012 | flow | NSE | 0.855 | **0.939** | 0.738 | 0.886 |
| | | LnNSE | 0.886 | **0.869** | 0.602 | 0.791 |
| | | CR | 0.811 | **0.803** | 0.852 | 0.811 |
| | SCF | RMSE | 0.158 | **0.150** | 0.165 | 0.150 |
| | | CSI | 0.985 | **0.999** | 0.960 | 0.992 |
| | | CR | 0.363 | **0.232** | 0.504 | 0.334 |
| 2013 | flow | NSE | 0.914 | 0.874 | **0.946** | 0.917 |
| | | LnNSE | 0.913 | 0.749 | **0.941** | 0.785 |
| | | CR | 0.679 | 0.605 | **0.827** | 0.619 |
| | SCF | RMSE | 0.053 | 0.063 | **0.049** | 0.055 |
| | | CSI | 0.992 | 0.987 | **0.994** | 0.990 |
| | | CR | 0.846 | 0.824 | **0.869** | 0.841 |
| 2014 | flow | NSE | 0.878 | 0.895 | 0.789 | **0.928** |
| | | LnNSE | 0.513 | 0.481 | 0.485 | **0.717** |
| | | CR | 0.627 | 0.627 | 0.712 | **0.647** |
| | SCF | RMSE | 0.079 | 0.087 | 0.078 | **0.078** |
| | | CSI | 0.996 | 0.993 | 0.990 | **0.996** |
| | | CR | 0.681 | 0.625 | 0.743 | **0.658** |
| No. acceptable models | | | 726 | 988 | 2245 | 604 |



**Table 7.** Cross-validation of streamflow and SCF predictions after relaxing the LOA criteria.

| Valid. year | Obs. | Likelihood measure | Calibration year | | | |
|---|---|---|---|---|---|---|
| | | | 2011 | 2012 | 2013 | 2014 |
| 2011 | flow | NSE | **0.881** | 0.861 | 0.769 | 0.854 |
| | | LnNSE | **0.839** | 0.838 | 0.711 | 0.796 |
| | | CR | **0.712** | 0.726 | 0.759 | 0.748 |
| | SCF | RMSE | **0.140** | 0.145 | 0.152 | 0.142 |
| | | CSI | **0.983** | 0.987 | 0.959 | 0.985 |
| | | CR | **0.551** | 0.450 | 0.615 | 0.552 |
| 2012 | flow | NSE | 0.808 | **0.914** | 0.758 | 0.837 |
| | | LnNSE | 0.822 | **0.918** | 0.595 | 0.791 |
| | | CR | 0.797 | **0.833** | 0.866 | 0.852 |
| | SCF | RMSE | 0.162 | **0.150** | 0.161 | 0.153 |
| | | CSI | 0.970 | **0.995** | 0.963 | 0.986 |
| | | CR | 0.417 | **0.342** | 0.516 | 0.439 |
| 2013 | flow | NSE | 0.947 | 0.896 | **0.940** | 0.941 |
| | | LnNSE | 0.940 | 0.880 | **0.934** | 0.914 |
| | | CR | 0.767 | 0.707 | **0.825** | 0.800 |
| | SCF | RMSE | 0.049 | 0.057 | **0.051** | 0.052 |
| | | CSI | 0.994 | 0.989 | **0.992** | 0.991 |
| | | CR | 0.857 | 0.843 | **0.871** | 0.862 |
| 2014 | flow | NSE | 0.872 | 0.859 | 0.787 | **0.898** |
| | | LnNSE | 0.540 | 0.307 | 0.310 | **0.674** |
| | | CR | 0.641 | 0.627 | 0.704 | **0.671** |
| | SCF | RMSE | 0.077 | 0.082 | 0.079 | **0.078** |
| | | CSI | 0.994 | 0.994 | 0.989 | **0.995** |
| | | CR | 0.706 | 0.661 | 0.748 | **0.713** |
| No. acceptable models | | | 419 | 813 | 2213 | 1029 |

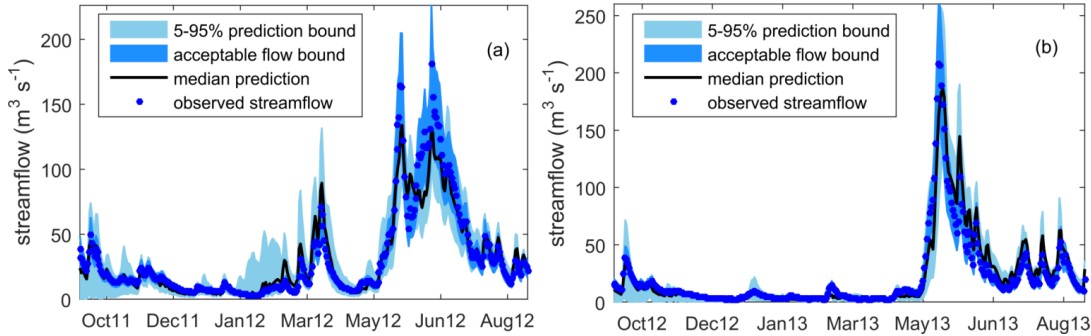

**Figure 7**. Prediction and acceptable flow bounds for sample calibration period (a) and validation period (b)




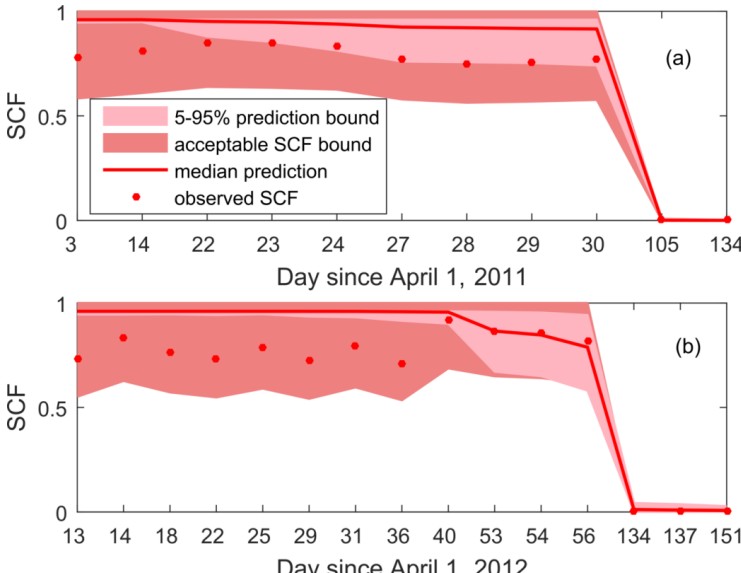

**Figure 8**. Prediction and acceptable bounds of average SCF for sample calibration period (a) and validation period (b)

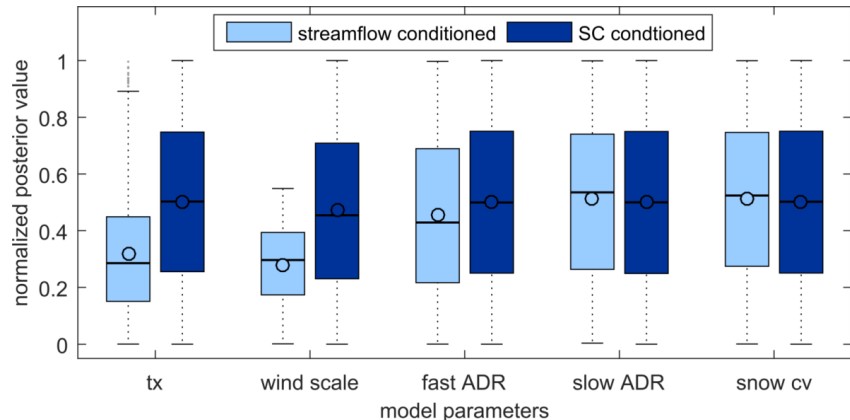

**Figure 9.** Boxplot showing posterior distribution of model parameters when separately conditioned using streamflow and

SCF. Parameter values are scaled between 0 and 1.





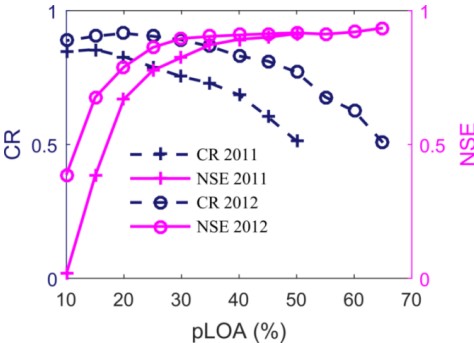

**Figure 10**. The effect of the percentage of observations required to be bracketed by each model realization (pLOA) on prediction uncertainty (CR) and efficiency of the median prediction (NSE) for sample calibration periods (years 2011 and 2012).

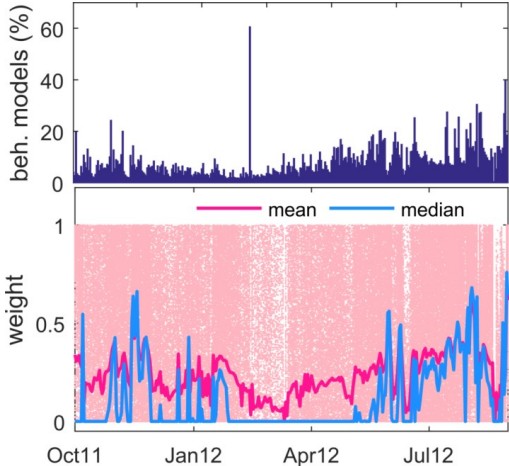

**Figure 11.** Daily percentage of acceptable model realizations with their predictions falling within the observation error bounds (upper plot) and the daily weight associated with each acceptable model realization as well as daily mean and median value of the weights (lower plot) in a sample calibration period.