# Peer review of "Parameter uncertainty analysis for an operational hydrological model using residual based and limits of acceptability approaches"

_Hydrology and Earth System Sciences, 2018_

## Referee Comment (RC1) · Anonymous Referee #1 · 6 Jun 2018

Review of "Parameter uncertainty analysis for an operational hydrological model using residual based and limits of acceptability approaches"

The manuscript is well-written and in line with the scope of this journal. It targets three different objectives: (1) uncertainty quantification / parameter estimation applied to an operation hydrological model; (2) investigation of the impact of using additional data to the output of the parameter estimation procedure; (3) assessment of using a time-relaxed (instead of limits-relaxed) GLUE LOA approach.

The approach is technically sound. Many aspects are addressed in a practical way, based on best-practices or heuristic approaches, leaving room for future more

theoretically-oriented investigations. The approach here is fairly justified by the objective of applying the methodology to a real-case scenario.

The conclusions are supported by an adequate number of tables and figures. As highlighted in one comment below, I think that the authors should provide more insights related to the interpretation of these results.

For these reasons, I would recommend to accept it with minor revision.

Detailed comments:

1 - Introduction: page 1, line 23: add reference or short inline explanation about the context for those readers who are not familiar with this company

2.1 - The hydrological model page 5, line 3: please add more information about the rationale behind the choice of these seven parameters and the corresponding low/high bounds.

2.2 - Study area and data I would suggest to add a figure depicting a map of the study area

3 - Results Figure 4 is not well readable in my opinion, because the variables are misaligned. An option could be to build a grid of subplots, leaving axis labels outside the grid and reporting the scatter plots of interest on the upper-diagonal cells and, for example, correlation values on the lower-diagonal. I leave the final decision to the authors. Page 10, line 13: why the validation results pertaining to year 2014 was not included in the corresponding figure 6? Please also elaborate on what are the possible motivations behind the poor performance of the behavioral models evaluated using LnNSE in year 2014.

---

## Referee Comment (RC2) · Anonymous Referee #2 · 19 Jun 2018

The manuscript is in general well-written and in line with the scope of HESS.

The manuscript performs a parameter uncertainty analysis of a distributed hydrological model by means of two variants of GLUE. Specifically the authors apply one variant based on the residuals and other in the limits of acceptability. The paper has a good practical component. However, I think it can be potentially highly improved by refining with a more rigorous and formal perspective.

Here i provide some comments that can I think can help to strength the paper:

1. introduction Pag 1 Line 23: there is not reference to Statkraft model Line 27: 'equally good' needs some definition or explanation Line 32: 'less justifiable' I would advice to

be more rigurous Pag 2 Line 8: 'Uncertainty analysis classification'. more references are needed

In general, i think, introduction is missing references and explanations and more rigour in the scientific writing I think the introduction would be benefit from these.

2. Methods and materials Pag 3 Line 30 and first paragraph: if possible, provide more explanation to define PT_GS_K Pag 5. Line 20: variables? Line 35: in my opinion the reference to the link fits better at the end of the paper, together with the date when the authors acceded to the data - this is for readers being able to reproduce the study 2.3.1. this section needs to provide more description, i.e. to guide the reader 2.3.2. equation (5): if possible, i suggest to use other letter different to L, as it can be confuse a reader to the likelihood function (i.e. in statistics the likelihood function is L(theta_model|data)) I suggest to use a more formal notation

In general, i think section 2 would be improved by 1) making clear distinction about statistic language from what the GLUE approach uses - which is what this paper uses. I think this might help to a reader that is not familiar with the approach to use it in right way with no confusion. 2) providing a road map of the methodology to re reader - for repeatability purposes

3. Results and 4 Discussion Same comment as in my last paragraph, I suggest to avoid confusion, when possible. for example, in pag 13, line 38 'information content'. This should be used with caution, as it requires an approach to be evaluated.

Finally, in case the editor ask to the authors a reviewed manuscript, i am happy to review it again

Kind Regards, Reviewer

---

## Author Comment (AC1) · 5 Jul 2018

**Response to Reviewer #1**

Dear Reviewer, we are grateful for your thoughtful comments and suggestions. Following is our reply to the points raised in your feedback; and it is structured as comment from reviewer (bold text) followed by our response to the comment. Where relevant, the specific changes are also highlighted.

**Review of "Parameter uncertainty analysis for an operational hydrological model using residual based and limits of acceptability approaches"**
**The manuscript is well-written and in line with the scope of this journal. It targets three different objectives: (1) uncertainty quantification / parameter estimation applied to an operation hydrological model; (2) investigation of the impact of using additional data to the output of the parameter estimation procedure; (3) assessment of using a timerelaxed (instead of limits-relaxed) GLUE LOA approach. The approach is technically sound. Many aspects are addressed in a practical way, based on best-practices or heuristic approaches, leaving room for future more theoretically-oriented investigations. The approach here is fairly justified by the objective of applying the methodology to a real-case scenario.**
**The conclusions are supported by an adequate number of tables and figures. As highlighted in one comment below, I think that the authors should provide more insights related to the interpretation of these results.**
**For these reasons, I would recommend to accept it with minor revision.**

As presented in our response to the following specific comments; additional references, explanations and figures have been provided in order to improve the manuscript through better readability and through the provision of additional insights into the work presented in the manuscript.

**1 - Introduction: page 1, line 23: add reference or short inline explanation about the context for those readers who are not familiar with this company**

A citation within the text has been added as follows:

Statkraft (2018) has recently released a new modelling framework mainly tailored for an operational purpose.

The following full reference has been added in the list of references section:
Statkraft information page: https://www.statkraft.com/, last access: 20 June 2018

**2.1 - The hydrological model page 5, line 3: please add more information about the rationale behind the choice of these seven parameters and the corresponding low/high bounds.**

Additional information has been provided as follows

In the default parameter settings of the PT_GS_K model seven parameters are considered as influential and thus allowed to vary in conditioning the model. Preliminary model calibration using the BOBYQA algorithm (Powell, 2009) and the default setting gave reasonable model performance. Hence, the same setting was also followed in this study with the addition that the sub-grid snow coefficient of variation was also considered an uncertain model parameter. A similar result was also observed when this setting was latter verified using the method of Morris (Morris, 1991; Saltelli et al., 2008) for screening the most influential out of the relevant model parameters.

The following information has been added in section 2.3.1 regarding the low/high bounds of the selected model parameters,

The feasible ranges of parameter values are set based on relevant literature and previous modelling studies in the Nea-Nidelva catchment.

**2.2 - Study area and data I would suggest to add a figure depicting a map of the study area**

The following physiographic and location map of the study area (Figure 1) has been added in the 'Study area and data' section. The figure number of the other figures and references to the figures in the manuscript has been updated accordingly.

[Figure]

**Figure 1**. Physiographic and location map of the Nea-catchment in Norway

**3 - Results Figure 4 is not well readable in my opinion, because the variables are misaligned. An option could be to build a grid of subplots, leaving axis labels outside the grid and reporting the scatter plots of interest on the upper-diagonal cells and, for example, correlation values on the lower-diagonal. I leave the final decision to the authors.**

Sub-plot labels are now placed in the diagonal cells, the scatter plots above the diagonals and the new cells with correlation coefficient scores have been added below the diagonals. A reference to the correlation coefficient scores is also included in page 9, line 32 as follows.

The aforementioned less sensitive model parameters can, however, have high effect on model outputs through interaction with other parameters. Some degree of interaction between model parameters can be seen from the correlation shown in Fig. 4. For example, a general decreasing trend in model performance can be noticed with a joint increase in *c1* and *c2*. The strong influence of *tx* in constraining the output is also evident in these plots. A considerable level of interaction can also be observed from the correlation coefficient scores between *c1* and *c2* (0.56), *c2* and *c3* (0.53) as well as between *tx* and *wind scale* (0.66).

Figure 4 and its caption have been modified as shown below:

[Figure]

**Figure 4**. Model performance in response to the interaction between model parameters (upper diagonal cells) and correlation coefficient scores between the parameters (lower diagonal cells)

**Page 10, line 13: why the validation results pertaining to year 2014 was not included in the corresponding figure 6? Please also elaborate on what are the possible motivations behind the poor performance of the behavioral models evaluated using LnNSE in year 2014.**

Plots depicting the validation result of using observations from year 2014, both when model parameters are identified using NSE alone (Fig. 6c) and when using combined likelihood (Fig. 6d), have been included in Figure 6. The following explanation was also added in page 10, line 14, regarding the

possible reasons for the observed poor performance when behavioural models are evaluated using LnNSE in year 2014:

This can be attributed to the relatively low quality of streamflow observations during the low streamflow period of this year. The validation result was also highly affected by nature of the likelihood measure used during the identification of behavioural models. For example, a persistent low performance was observed during early months of the hydrologic year when validating model parameters identified using NSE alone (Fig. 6c) as compared to those identified using the combined likelihood (Fig. 6d). Similarly, excluding the first 30 observations from the validation dataset resulted to an improvement of LnNSE from -0.53 to 0.44.

[Figure]

**Figure 6**. Median, 5-95 percentile range and observed values of streamflow for sample calibration period (a) and validation periods (b, c and d). The calibration result (a) as well as the validation results presented in (b) and (d) are based on behavioural models identified using the combined likelihood, while the result shown in (c) is based on behavioural models identified using NSE alone.

**New references:**

Beven, K. J., Smith, P. J., and Freer, J. E.: So just why would a modeller choose to be incoherent?, Journal of hydrology, 354, 15-32, 2008.

Clark, M. P., Kavetski, D., and Fenicia, F.: Pursuing the method of multiple working hypotheses for hydrological modeling, Water Resources Research, 47, 2011.

Copernicus land monitoring service: https://land.copernicus.eu/pan-european/corine-land-cover, accessed on: 29 August 2016

Mantovan, P., and Todini, E.: Hydrological forecasting uncertainty assessment: Incoherence of the GLUE methodology, Journal of hydrology, 330, 368-381, 2006.

Morris, M. D.: Factorial sampling plans for preliminary computational experiments, Technometrics, 33, 161-174, 1991.

Norwegian mapping authority: https://www.kartverket.no/, accessed on: 1 September 2016

Pianosi, F., Beven, K., Freer, J., Hall, J. W., Rougier, J., Stephenson, D. B., and Wagener, T.: Sensitivity analysis of environmental models: A systematic review with practical workflow, Environmental Modelling & Software, 79, 214-232, 2016.

Powell, M. J.: The BOBYQA algorithm for bound constrained optimization without derivatives, Cambridge NA Report NA2009/06, University of Cambridge, Cambridge, 26-46, 2009.

Saltelli, A., Ratto, M., Andres, T., Campolongo, F., Cariboni, J., Gatelli, D., Saisana, M., and Tarantola, S.: Global sensitivity analysis: the primer, John Wiley & Sons, 2008.

Statkraft information page: https://www.statkraft.com/, last access: 20 June 2018

Stedinger, J. R., Vogel, R. M., Lee, S. U., and Batchelder, R.: Appraisal of the generalized likelihood uncertainty estimation (GLUE) method, Water resources research, 44, 2008.

Wagener, T., McIntyre, N., Lees, M., Wheater, H., and Gupta, H.: Towards reduced uncertainty in conceptual rainfall-runoff modelling: Dynamic identifiability analysis, Hydrological Processes, 17, 455-476, 2003.

---

## Author Comment (AC2) · 5 Jul 2018

**Response to Reviewer #2**

Dear Reviewer, we are grateful for your thoughtful comments and suggestions. Following is our reply to the points raised in your feedback; and it is structured as comment from reviewer (bold text) followed by our response to the comment. Where relevant, the specific changes are also highlighted.

**The manuscript is in general well-written and in line with the scope of HESS. The manuscript performs a parameter uncertainty analysis of a distributed hydrological model by means of two variants of GLUE. Specifically the authors apply one variant based on the residuals and other in the limits of acceptability. The paper has a good practical component. However, I think it can be potentially highly improved by refining with a more rigorous and formal perspective.**

As presented in our response to the following specific comments; additional explanations and references have been provided and some corrections are made to make the manuscript more rigorous and to improve its readability.

**1. introduction**
**Page 1 Line 23: there is not reference to Statkraft**

A citation within the text has been added as follows:

Statkraft (2018) has recently released a new modelling framework mainly tailored for an operational purpose.

The following full reference has been added in the list of references section:
Statkraft information page: https://www.statkraft.com/, last access: 20 June 2018

**Page 1 Line 27: 'equally good' needs some definition or explanation**

An explanation and additional reference have been provided as follows:

During calibration, equifinality arises when different parameter sets give equally good results in terms of predefined efficiency criteria (Beven, 1993; Savenije, 2001; Wagener et al, 2003).

**Page 1 Line 32: 'less justifiable' I would advice to be more rigorous**

The text has been modified and an explanation added from another reference as follows:

It is based on the concept that all models of hydrological systems are highly simplified representations of reality (e.g. Reichert and Omlin, 1997), and hence it is expected to have several different model structures and parameter sets that describe the system in an adequate way (Blazkova and Beven, 2002). When dealing with non-linear systems, the classic hydrological approach of using a single set of model parameters may lead to large predictive biases (e.g. Mantovan and Todini, 2006).

**Page 2 Line 8: 'Uncertainty analysis classification'. more references are needed**

As suggested, more references have been added to this paragraph.

Uncertainty analysis techniques can be classified as frequentist or Bayesian approaches, probabilistic or non-probabilistic approaches (e.g. Montanari et al. 2009), or as formal or informal approaches (e.g. Vrugt et al., 2009). Among the most widely used techniques in hydrological modelling are the formal

Bayesian and the GLUE methods (Jin et al., 2010). The formal Bayesian approach makes strong assumptions about the statistics of observed data; with the likelihood function defined based on assumptions about the nature of the residuals (Beven et al, 2008; Schoups and Vrugt, 2010). However, the choice of an adequate likelihood function has been the subject of considerable debate. According to Beven and Smith (2014), a formal probabilistic likelihood function will have limited value since non-stationary epistemic uncertainties cannot be adequately represented by a statistical model. In GLUE, the likelihood measure is associated with a parameter set and should ideally reflect all the different sources of uncertainty (Beven and Smith, 2014). The original GLUE methodology has been subject of debate for using informal likelihoods and a subjectively set threshold of behavioral models (e.g. Mantovan and Todini, 2006; Stedinger et al., 2008; Clark et al., 2011; Nearing et al., 2016). This problem is common to most residual-based model selection methods (Schaefli, 2016). The extended concept of behavioral models in the GLUE limits of acceptability approach (GLUE LOA) (Beven, 2006) attempts to overcome this drawback through use of error bounds of the observational dataset.

**2. Methods and materials**
**Page 3 Line 30 and first paragraph: if possible, provide more explanation to define PT_GS_K**

Additional explanation has been provided as shown in the highlighted text.

The Statkraft Hydrological Forecasting Toolbox, Shyft, (https://github.com/statkraft/shyft) is an open-source distributed hydrological modelling framework developed by Statkraft (Burkhart et al., 2016). The modelling framework has three main models (method stacks) and in this study, one of these models, PT_GS_K, was used for uncertainty analysis. PT_GS_K is a conceptual model with several adjustable parameters depending on the climatic and physiographic characteristics of the study area where the model is applied. This model requires temperature, precipitation, radiation, relative humidity, and wind speed as forcing data. PT_GS_K uses the Priestley-Taylor (PT) method (Priestley, 1972) for estimating potential evaporation; a quasi-physical based method for snow melt, sub-grid snow distribution and mass balance calculations (GS method); and a simple storage-discharge function (Lambert, 1972; Kirchner, 2009) for catchment response calculation (K). Overall, these three methods constitute the PT_GS_K model in Shyft. The framework establishes a sequence of spatially distributed cells of arbitrary size and shape. As such it can provide lumped (single cell) or discretized (spatially distributed) calculations, as in this study. The model was applied to each of the grid-cells and for each time step.

**Page 5 Line 20: variables?**

'parameter' has been changed to 'variable' and a previous reference to the forcing variables was also indicated as shown below:

As mentioned in section 2.1, the PT_GS_K model requires temperature, precipitation, radiation, relative humidity, and wind speed as forcing data. In this study, daily time series data of these variables for the study area were obtained from Statkraft (2018) as point measurement, with the exception of relative humidity.

**Page 5 Line 35: in my opinion the reference to the link fits better at the end of the paper, together with the date when the authors acceded to the data - this is for readers being able to reproduce the study**

A text citation has been added to the manuscript and the reference to the link is moved to the reference section at the end of the manuscript. The same change was also made for the reference related to the DEM data from Norwegian Mapping Authority.

Data for these physiographic variables was retrieved from two sources: the land cover data from Copernicus land monitoring service (2016) and the 10m digital elevation model (10m DEM) from the Norwegian mapping authority (2016).

**2.3.1. this section needs to provide more description, i.e. to guide the reader**

Additional description as well as relevant references have been added as shown below:

The performance of all uncertainty analysis techniques depends on the efficiency of the sample to represent the entire response surface (Pappenberger et al., 2008). In this study, prior distributions of the uncertain model parameters were not known and hence a uniform distribution was assumed. The challenge in using uniform distribution is, however, to adequately sample the entire parameter dimensions. To overcome this challenge and to better identify regions of behavioral simulations, a sample size of 100,000 runs was used. Each model run is a realization of a parameter set randomly drawn from the domains of the model parameters. The feasible ranges of parameter values are set based on information from relevant literature and previous modelling studies in the Nea-Nidelva catchment. An All-At-a-Time (AAT) sampling method (Pianosi et al., 2016) was employed. This method involves random selection of all parameter values simultaneously. Matlab scripts from the SAFE toolbox (Pianosi et al., 2015) were used as a basis to characterize behavioral and non-behavioral models.

**2.3.2. equation (5): if possible, i suggest to use other letter different to L, as it can confuse a reader to the likelihood function (i.e. in statistics the likelihood function is L(theta_model|data)). I suggest to use a more formal notation**

The notation for the Nash-Sutcliffe efficiency based informal likelihood measure has been changed from $L$ to $L_{NS}$ as follows:

$$L_{NS}\big(O|M(\theta_i)\big) = 0.46(L_{LnNSE}) + 0.54(L_{NSE}) \tag{4}$$

**In general, i think section 2 would be improved by 1) making clear distinction about statistic language from what the GLUE approach uses - which is what this paper uses. I think this might help to a reader that is not familiar with the approach to use it in right way with no confusion. 2) providing a road map of the methodology to re reader – for repeatability purposes**

Although there are some terminologies in the manuscript that are used both by the formal statistical and informal (GLUE) uncertainty analysis approaches, we are unable to identify a language in the manuscript which was not used in previous GLUE related studies.

Regarding the second point, as shown above, more description has been provided in section 2.3.1

**3. Results and 4 Discussion Same comment as in my last paragraph, I suggest to avoid confusion, when possible. for example, in page 13, line 38 'information content'. This should be used with caution, as it requires an approach to be evaluated.**

Additional explanation and a reference have been provided to describe the context in which this terminology was used:

The information content of the input and observation datasets, which in turn is a function of multiple factors, such as relative accuracy and unusualness of an event (Beven and Smith, 2015) is more important than the length of the datasets especially in continuous rainfall-runoff modelling.

**New references:**

Beven, K. J., Smith, P. J., and Freer, J. E.: So just why would a modeller choose to be incoherent?, Journal of hydrology, 354, 15-32, 2008.

Clark, M. P., Kavetski, D., and Fenicia, F.: Pursuing the method of multiple working hypotheses for hydrological modeling, Water Resources Research, 47, 2011.

Copernicus land monitoring service: https://land.copernicus.eu/pan-european/corine-land-cover, accessed on: 29 August 2016

Mantovan, P., and Todini, E.: Hydrological forecasting uncertainty assessment: Incoherence of the GLUE methodology, Journal of hydrology, 330, 368-381, 2006.

Morris, M. D.: Factorial sampling plans for preliminary computational experiments, Technometrics, 33, 161-174, 1991.

Norwegian mapping authority: https://www.kartverket.no/, accessed on: 1 September 2016

Pianosi, F., Beven, K., Freer, J., Hall, J. W., Rougier, J., Stephenson, D. B., and Wagener, T.: Sensitivity analysis of environmental models: A systematic review with practical workflow, Environmental Modelling & Software, 79, 214-232, 2016.

Powell, M. J.: The BOBYQA algorithm for bound constrained optimization without derivatives, Cambridge NA Report NA2009/06, University of Cambridge, Cambridge, 26-46, 2009.

Saltelli, A., Ratto, M., Andres, T., Campolongo, F., Cariboni, J., Gatelli, D., Saisana, M., and Tarantola, S.: Global sensitivity analysis: the primer, John Wiley & Sons, 2008.

Statkraft information page: https://www.statkraft.com/, last access: 20 June 2018

Stedinger, J. R., Vogel, R. M., Lee, S. U., and Batchelder, R.: Appraisal of the generalized likelihood uncertainty estimation (GLUE) method, Water resources research, 44, 2008.

Wagener, T., McIntyre, N., Lees, M., Wheater, H., and Gupta, H.: Towards reduced uncertainty in conceptual rainfall-runoff modelling: Dynamic identifiability analysis, Hydrological Processes, 17, 455-476, 2003.

---

## Author Response (AR1)

Dear Reviewers, we are grateful for your thoughtful comments and suggestions. Following is our reply to the points raised in your feedback; and it is structured as comment from reviewer (bold text) followed by our response to the comment. The specific changes are shown in the marked-up version of the manuscript following the reply to comments section.

**Response to Reviewer #1**

**Review of "Parameter uncertainty analysis for an operational hydrological model using residual based and limits of acceptability approaches"**
**The manuscript is well-written and in line with the scope of this journal. It targets three different objectives: (1) uncertainty quantification / parameter estimation applied to an operation hydrological model; (2) investigation of the impact of using additional data to the output of the parameter estimation procedure; (3) assessment of using a time relaxed (instead of limits-relaxed) GLUE LOA approach. The approach is technically sound. Many aspects are addressed in a practical way, based on best-practices or heuristic approaches, leaving room for future more theoretically-oriented investigations. The approach here is fairly justified by the objective of applying the methodology to a real-case scenario.**
**The conclusions are supported by an adequate number of tables and figures. As highlighted in one comment below, I think that the authors should provide more insights related to the interpretation of these results.**
**For these reasons, I would recommend to accept it with minor revision.**

Thank you, as presented in our response to the following specific comments; additional references, explanations and figures have been provided in the revised manuscript.

**1 - Introduction: page 1, line 23: add reference or short inline explanation about the context for those readers who are not familiar with this company**

A reference has been added in the revised manuscript.

**2.1 - The hydrological model page 5, line 3: please add more information about the rationale behind the choice of these seven parameters and the corresponding low/high bounds.**

Additional information has been provided about the rationale behind the choice of these parameters and the corresponding low/high bounds.

**2.2 - Study area and data I would suggest to add a figure depicting a map of the study area**

Physiographic and location map of the study area (Figure 1) has been added in the 'Study area and data' section. The figure number of the other figures and references to the figures in the manuscript has been updated accordingly.

**3 - Results Figure 4 is not well readable in my opinion, because the variables are misaligned. An option could be to build a grid of subplots, leaving axis labels outside the grid and reporting the scatter plots of interest on the upper-diagonal cells and, for example, correlation values on the lower-diagonal. I leave the final decision to the authors.**

Sub-plot labels are now placed in the diagonal cells, the scatter plots above the diagonals and the new cells with correlation coefficient scores have been added below the diagonals. A reference to the correlation coefficient scores is also included in page 10, line 6.

**Page 10, line 13: why the validation results pertaining to year 2014 was not included in the corresponding figure 6? Please also elaborate on what are the possible motivations behind the poor performance of the behavioral models evaluated using LnNSE in year 2014.**

Plots depicting the validation result of using observations from year 2014, both when model parameters are identified using NSE alone (Fig. 7c) and when using combined likelihood (Fig. 7d), have been included in Figure 7 (Figure 6 in the original manuscript). An explanation was also added in page 10, line 26, regarding the possible reasons for the observed poor performance when behavioural models are evaluated using LnNSE in year 2014:

**Response to Reviewer #2**

**The manuscript is in general well-written and in line with the scope of HESS. The manuscript performs a parameter uncertainty analysis of a distributed hydrological model by means of two variants of GLUE. Specifically the authors apply one variant based on the residuals and other in the limits of acceptability. The paper has a good practical component. However, I think it can be potentially highly improved by refining with a more rigorous and formal perspective.**

Thank you, as presented in our response to the following specific comments; additional explanations and references have been provided and some corrections are made in the revised manuscript.

**1. introduction**
**Page 1 Line 23: there is not reference to Statkraft**

A reference has been added.

**Page 1 Line 27: 'equally good' needs some definition or explanation**

An explanation and additional reference have been provided in the revised manuscript.

**Page 1 Line 32: 'less justifiable' I would advice to be more rigorous**

The text has been modified and an explanation was also added from another reference.

**Page 2 Line 8: 'Uncertainty analysis classification'. more references are needed**

As suggested, more references have been added to this section.

**2. Methods and materials**
**Page 3 Line 30 and first paragraph: if possible, provide more explanation to define PT_GS_K**

Additional explanation on PT_GS_K has been provided in the revised manuscript.

**Page 5 Line 20: variables?**

'parameter' has been changed to 'variable' in the revised manuscript.

**Page 5 Line 35: in my opinion the reference to the link fits better at the end of the paper, together with the date when the authors acceded to the data - this is for readers being able to reproduce the study**

A text citation has been added to the manuscript and the reference to the link is moved to the reference section at the end of the manuscript. The same change was also made for the reference related to the DEM data from Norwegian Mapping Authority.

**2.3.1. this section needs to provide more description, i.e. to guide the reader**

Additional description as well as relevant references have been added to this section.

**2.3.2. equation (5): if possible, i suggest to use other letter different to L, as it can confuse a reader to the likelihood function (i.e. in statistics the likelihood function is L(theta_model|data)). I suggest to use a more formal notation**

The notation for the informal likelihood based on the Nash-Sutcliffe efficiency has been changed from $L$ to $L_{NS}$

**In general, i think section 2 would be improved by 1) making clear distinction about statistic language from what the GLUE approach uses - which is what this paper uses. I think this might help to a reader that is not familiar with the approach to use it in right way with no confusion. 2) providing a road map of the methodology to re reader – for repeatability purposes**

Although there are some terminologies in the manuscript that are used both by the formal statistical and informal (GLUE) uncertainty analysis approaches, we are unable to identify a technical word in the manuscript which was not used in previous GLUE related studies.

Regarding the second point, as shown above, more description has been provided in section 2.3.1

**3. Results and 4 Discussion Same comment as in my last paragraph, I suggest to avoid confusion, when possible. for example, in page 13, line 38 'information content'. This should be used with caution, as it requires an approach to be evaluated.**

The terminology: 'information content' was used here in a general context to represent the value of a dataset in constraining model parameters. The sentence has been rephrased and a reference is provided in the revised manuscript to clarify the context in which this terminology was used. If it still appears confusing, we can remove the sentence.

This terminology has been removed from section 3.1.4 in the revised manuscript.

**New references:**

[revised manuscript text omitted]